# Occupational Health: Physical Activity, Musculoskeletal Symptoms and Quality of Life in Computer Workers: A Narrative Review

**DOI:** 10.3390/healthcare10122457

**Published:** 2022-12-05

**Authors:** Sara Moreira, Maria Begoña Criado, Paula Clara Santos, Maria Salomé Ferreira, Carla Gonçalves, Jorge Machado

**Affiliations:** 1ICBAS, Abel Salazar Institute for Biomedical Sciences, University of Porto 4099-002 Porto, Portugal; 2ESS IPVC, School of Health, Polytechnic Institute of Viana do Castelo, 4900-314 Viana do Castelo, Portugal; 3CBSin—Center of BioSciences in Integrative Health, 4000-105 Porto, Portugal; 4TOXRUN—Toxicology Research Unit, University Institute of Health Sciences, CESPU, 4585-116 Gandra, Portugal; 5ESS PPorto—Department of Physiotherapy, School of Health, Polytechnic of Porto, 4200-072 Porto, Portugal; 6CIAFEL—Research Center in Physical Activity, Health and Leisure, Faculty of Sport, University of Porto (FADEUP) and Laboratory for Integrative and Translational Research in Population Health (ITR), 4200-450 Porto, Portugal; 7CIR—Centro de Investigação e Reabilitação, School of Health, Polytechnic of Porto, 4200-072 Porto, Portugal; 8UICISA, E—Health Sciences Research Unit, Nursing, Nursing School of Coimbra (ESEnfC), Portugal School of Health, Polytechnic Institute of Viana do Castelo, 4900-314 Viana do Castelo, Portugal; 9ESDL IPVC, School of Sport and Leisure, Polytechnic Institute of Viana do Castelo, 4900-347 Viana do Castelo, Portugal; 10SPRINT—Research Center in Sports Performance, Recreation, Innovation and Technology, 4960-320 Melgaço, Portugal; 11LABIOMEP—Laboratório de Biomecânica do Porto, Universidade do Porto, 4200-450 Porto, Portugal

**Keywords:** quality of life, occupational health, musculoskeletal symptoms, physical activity level, computer workers

## Abstract

Computer work has assumed a very important role for many companies, but specific occupational and health symptoms associated with the use of computers can appear. According to the literature, physical activity is considered a key player in the prevention and control of work-related musculoskeletal symptoms, contributing to a better quality of life in computer workers. The principal aims of this review are to contribute to a better understanding of the relationship among sedentary behavior, physical activity and quality of life of computer workers; to outline the importance of promoting the development of an educational program for Occupational Health directed at computer workers at the level of quality of life, musculoskeletal symptoms, and physical activity; and to call for attention to the factors needed to initiate and maintain a health program that involves the active participation of workers who must be concerned about their health. This review demonstrates the important role of workplaces in health promotion and the opportunities that workplaces provide to establish and continue successful health-promotion programs for computer workers, generating important benefits for the health and quality of life of individuals and groups.

## 1. Introduction

Occupational Health consists of an area of intervention that values the workplace as a privileged space for the prevention of professional risks, health protection and promotion, and access to health and safety services at work [1]. According to the World Health Organization (WHO), the main purpose of Occupational Health Service is to promote working conditions that guarantee the highest degree of quality of life at work; protect the health of workers; promote physical, mental and social well-being; and prevent and control accidents and diseases by reducing risk factors [2].

Currently, new technologies are revolutionizing contemporary social and working life by replacing physically demanding tasks with essentially sedentary ones [3,4]. With technological evolution there has been a change in the professional practice of workers, with an increase in the use of computers and new communication and information technologies. Computer work has assumed a very important role in the daily lives of many companies [5]. The use of digital technologies, particularly computers, has led to an improvement in workplace conditions and also to the presence of specific health problems that frequently affect the musculoskeletal system [6,7,8].

In this context, the European Agency for Safety and Health at Work [9] has recently promoted a campaign for Healthy Workplaces 2023–2025 entitled “Safe and healthy work in the digital age,” with the main objective of raising awareness of the impact of new digital technologies on work and workplaces and the challenges and opportunities associated with occupational safety and health.

Office or computer workers (CWS) are considered those who spend most of their working time seated and using computers. This sedentary position it is thought to increase the risk of developing different chronic diseases [10,11].

Depending on the profession, musculoskeletal symptoms (MSS) can occur in several places directly related to the function performed and the associated technical gesture. By maintaining an evolutionary framework of increasingly mechanistic work it is possible to predict that physical factors such as the overload of some muscle groups, stress, prolonged maintenance of incorrect postures, repeatability of the same movement pattern, and the mechanical compression of body structures are important risk factors for the emergence of MSS. When MSS is caused or aggravated by professional activity in an occupational environment, it is called work-related musculoskeletal disorder [12].

According to the European Risk Report by the European Agency for Safety and Health at Work, other important factors that can influence the appearance of work-related musculoskeletal disorders are (i) individual factors, such as low physical fitness, age (the elderly are at greater risk) and obesity; (ii) environmental factors, such as environments with inadequate lighting, temperatures, noise, and work environments not adapted to the needs of the worker; and (iii) organizational and psychosocial factors such as monotonous tasks and low job satisfaction [13]. In addition to the physical health repercussions that come from the nature of work, factors that contribute to the decrease in workers’ quality of life (QoL) include mental health influenced by stress, the relationship with managers and colleagues linked to a high level of demand, a high workload, and performance evaluation [14].

Several therapeutic approaches can be used to reduce MSS, among them physical activity (PA) [15]. According to the WHO [2], physical activity is any body movement produced by muscles that require energy expenditure. Nevertheless, what has been reported in the literature is that the majority of MSS patients do not perform physical activity in a regular way. Several reasons for this were given with a shortage of time, a number of working hours, and absence of motivation being the most reported [2,16,17].

An increase in PA allows for a reduction in total sitting time, leading to an improvement in physical and mental health status [18,19,20]. PA is considered a key player in the prevention and control of chronic, non-communicable diseases and in the fight against cognitive decline, anxiety, and depression [18,19,20,21].

Well-being at work is crucial for employee satisfaction and productivity, and increased physical activity could be a health promotion strategy with positive effects on quality of life (QoL). Quality of Life is defined by the WHO as an individual’s perception of their position in life within the context of the culture and value system in which they are inserted and in relation to their goals, expectations, standards, and concerns [22].

In December 2019, the WHO declared the COVID-19 pandemic state that remains to date. Therefore, in 2020, with a compulsory lockdown in place to prevent transmission of the SARS-CoV-2 coronavirus, a reorganization of companies was needed as homeworking had become mandatory for everyone [23]. The confinement led to profound lifestyle changes for the population with significant effects on health [24,25]. Restrictions on movement and bans on performing outdoor activities altered routines, reduced the levels of physical activity, and increased the likelihood of sedentary behavior, anxiety, and depression, all of which are health risk factors [26].

During confinement, homes became the place of work, education, and leisure for everyone. Although teleworking can bring some advantages including increased productivity, a lack of clarity surrounding the limits between home and work, excessive working hours and, in many cases, a lack of companies’ support can have a negative influence on the physical and mental health of workers [27], contributing to a decrease in their health and quality of life [27,28].

In this context, research on the effects of physical exercise on computer workers’ health [18,19,20,21] concluded that it has a positive influence on musculoskeletal pain symptoms, namely neck and lower back pain [20,21,29,30,31]. Additionally, studies performed during the pandemic suggested that regular physical activity can help the immune response [32].

Considering that physical activity is a useful strategy to reduce MSS, improving PA among computer workers will be an important challenge for healthcare. Therefore, the development of health-promotion actions in a work context will be an important opportunity to promote physical activity [19,24,29].

According to new guidelines from the World Health Organization [2], to obtain health benefits all adults between 18 and 64 years of age should perform regular physical activities [2]. PA of any duration is considered to be associated with improved health gains. These new WHO recommendations for adults specify a target range of 150 to 300 min of moderate-intensity physical activity and 75 to 150 min of vigorous-intensity physical activity per week [2]. With respect to computer workers, the study performed by Weenas et al. (2019) reported that they are less likely to comply with WHO guidelines when compared to other professionals [33].

In the context of Occupational Health, it will be important to focus on how to change workers’ attitudes in order to introduce adequate physical activity-promotion programs. To that end, the computer workers’ perception of PA and MSS must first be investigated [10,19,34]. At the same time, determining the most frequent musculoskeletal injuries and the level of PA in computer workers will be important for planning projects for the prevention of these conditions and for the future promotion of health.

According to previous research conducted by our group, in Portugal there are few studies on health promotion programs involving the assessment of PA, presence of MSS, and level of QoL [35,36,37,38] in computer workers. However, given the social and cultural characteristics of our country, these kinds of studies are essential for contributing to the development of successful actions that promote health in a work-context activity [19,22,29].

Taking this into consideration, the main goal of the present work is to review the literature to provide a deeper understanding of PA, musculoskeletal symptoms, and quality of life in the context of occupational health, and to review factors that could contribute to the design of successful health-oriented physical activity-promotion actions directed at computer workers.

## 2. Material and Methods

PubMed, Web of Science and The Cochrane Library databases were searched using the (MeSH) terms ‘computer workers,’ ‘physical activity,’ ‘musculoskeletal symptoms,’ ‘quality of life,’ and ‘occupational health.’ These terms were internally validated by the co-authors. There were several inclusion criteria for the articles: [1] they must have been published between 2000 and 2021; [2] they must have been available as a full text in English; and [3] they must have been original research, reviews, meta-analyses, or letters to the editor. If all inclusion criteria were met and titles and abstracts were verified, the articles were fully read. If a full text revealed that not all requirements were present, that paper was excluded. Additional literature was obtained by searching references in the manuscripts (the snowball method).

## 3. Literature Revision

### 3.1. Quality of Life & Occupational Health in Computer Workers

According to the World Health Organization (WHO), quality of life (QoL) is multidimensional and encompasses various aspects of an individual’s life in a biopsychosocial approach [1,39,40]. This concept is subjective and tends to change throughout life characterized by an intentional lifestyle choice. These differences can be seen between genders, age groups, educational levels, the presence or absence of adverse factors, professions, and lifestyles, among other aspects that affect individuals’ perceptions of their health status [11,39,41,42,43].

The concept of health can be seen and interpreted from an individual, social, and cultural perspective [44]. According to the WHO, health is seen as a resource for everyday life as a dimension of QoL [39,42,45]. Measuring the health statuses of populations makes it possible to define levels of comparison between groups regarding health conditions (due to different pathologies), working conditions (such as working hours, work situation, or leadership roles), social conditions, economic conditions, and even conditions such as sex and age [42].

The physical, mental, economic, and social well-being of workers is influenced by their work context which, when safe and healthy, will inevitably have a positive impact on their health and well-being as well as that of their employees, families, communities, and society at large [1].

Currently, computers are a key element in the daily work of many individuals, representing the most common type of work in Europe and involving millions of workers [6]. They also represent a potential risk to the workers’ health [7,8]. With the increasing affluence of technology, the use of computers in the workplace implies a constant repetition of movement and/or the adoption of the same posture for most of the day, promoting sedentary behaviors that directly affect QoL [46,47].

The strong predominance of sedentary behavior implies that CWS spend most of the time performing various tasks in a seated position. Maintaining such a posture without a rest interval leads to an excessive use of the musculoskeletal system, which can cause discomfort, fatigue, or trauma and cause a predisposition to develop disorders of the same nature [24,44,45,46,48,49].

Inadequate and flexion postures for long periods of work and repetitive tasks can produce excessive tension in muscular and ligament structures, increasing the musculoskeletal disorders associated with work [10,50,51,52], often the cervical and lumbar regions and usually at least one region of the upper limbs [7,8,14,20,50,53,54,55,56].

These working conditions, including a repetition of the work gesture and sedentary behavior, lead to a greater risk of developing chronic diseases and consequently the risk of decreasing the quality of life [10,11,57].

These workers are therefore at a greater risk of developing various chronic diseases due to the repetition of the work gesture and their sedentary behavior [10,11] and, consequently, they are at risk of decreasing their quality of life [57]. The literature reports that the prevalence of these types of repercussions is associated with the alteration of other mental health risk factors such as higher levels of anxiety, sleep problems, and general fatigue [43,48,49,58]. Intense exposure to the computer, short deadlines for delivery of work, high degree of responsibility, and a high mental load are also determining factors for the decrease in mental well-being. All of these are factors that influence a worker’s QoL [10,48,59]. According to the European Agency for Safety and Health at Work (2019), these risk factors are associated with the way in which work is carried out, organized, and managed as well as with the economic and social context, leading to a higher level of stress and causing a deterioration of physical and mental health [16,59].

The literature on this subject suggests that the participation of health professionals (such as ergonomists or physical therapists) in an occupational context has a positive effect on the prevention and reduction of work-related musculoskeletal disorders [60].

In this context, physiotherapy is considered in the literature as a cost-effective strategy to reduce the risk of work-related musculoskeletal disorders by means of the implementation of specific physical exercise programs. These programs can have several benefits for both the worker and the company, namely increasing productivity [30,60,61]. These exercise programs can be carried out in a work context with the aim of increasing the strength and flexibility of the most requested muscles during the workday, in addition to enhancing social integration and improving the quality of life of workers [30,48,62]. In this context, social–ecological models that describe the interactive characteristics of individuals and environments that underlie health outcomes have long been recommended to guide public health practice [63]. Thus, the role of a multidisciplinary team including a physical therapist becomes essential to improve the adhesion to physical activity and thereby prevent and/or treat musculoskeletal injuries and improve quality of life [61,62,63].

Holzgreve et al. (2018) reported that the implementation of a physical activity program in the workplace improved the range of motion, diminished musculoskeletal injuries, and resulted in an improvement in quality of life [14]. This kind of approach is easily accessible, allows the simultaneous participation of several workers, and improves postures with movement variability, stretching the shortened muscles of the sitting posture and mobilizing all joints.

In fact, one of the health-protective factors described in the literature is an active lifestyle associated with a diminished risk of chronic disease and an increase in QoL [10]. According to Mayer et al., any form of strength work, such as resistance exercises, has been shown to be a fundamental tool in both preventing and reducing pain symptoms [64].

With respect to computer workers, recent reviews on the effect of physical activity on health status [18,20,21] reported a significant and protective effect on pain symptoms, namely neck and lower back pain [20,29,30], and a significant association between physical activity and QoL [19,20,30,65]. These reviews focused primarily on computer workers with common chronic work-related disorders such as musculoskeletal pain, but there is little research on QoL and healthy computer workers that also addresses sedentary behavior over several working hours, putting computer workers at high risk of developing work-related chronic diseases, or QoL in computer workers with other types of work-related illnesses [29,48,65].

Improving QoL must be a priority in the framework of the workers’ health. To that end, creating health-promotion actions in the work context will be crucial [19,29,66]. To study the workers’ perception of QoL, it will be fundamental to know how to change attitudes and implement successful, health-oriented physical activity-promotion programs in different social groups [19,34,65,67,68].

### 3.2. Physical Activity & Occupational Health in Computer Workers

As was previously indicated, the labor demand of some professions in the 21st century compromises human movement by reducing the level of daily physical activity. According to Stockwell et al., PA is defined as a body movement produced by the musculoskeletal system resulting in energy expenditure in activities such as walking [69].

Computer workers perform precise movements using electronic devices, generating an increase in physical inactivity and a decrease in PA during working hours [17,19,70].

As was previously discussed, regular practice of physical activity is recommended to reduce MSS. However, what is reported in different studies indicates that most of the people who suffer from MSS do not perform PA in a regular way [2,16,17]. In this sense, most studies only analyze social/economic factors but, as Calonge Pascual et al. suggested, PA-related adherence factors must be studied with a multi-dimensional approach under the WHO’s five dimensions adherence model [71].

Recent studies carried out during the pandemic investigated the effects of physical exercise on the health of computer workers [18,20,29] and reported the benefits of physical exercise on symptoms of musculoskeletal pain (i.e., neck pain and lower back pain) [20,21,29,30]. Moderate and vigorous PA is associated with a decrease in C-reactive proteins and an increase in anti-inflammatory mediators, thus decreasing inflammation and consequently pain. PA also seems to provide a state of well-being that enables individuals to realize their own potential [68].

Improving PA with the aim of reducing MSS is a challenge, and the work context will play an important role in creating and developing health-promotion actions [19,29,66]. In this context, a multidisciplinary approach is necessary to investigate the perceptions that computer workers have about PA and MSS. This will help to change attitudes and promote successful, health-oriented physical activity programs [19,34].

In summary, scientific evidence demonstrates the benefits that PA has on health. Its practice should be encouraged and promoted for its inherent benefits. An increase in PA levels allows for a reduction in the total time spent sitting, leading to an improvement in the state of physical health and mental health and acting as an ally in the prevention and control of chronic, non-communicable diseases [18,19,33,56]. The promotion of physical activity and the adoption of healthy lifestyles in an occupational context is crucial for individual and community health gains [19,39].

### 3.3. Musculoskeletal Symptomatology & Occupational Health

The incidence of work-related musculoskeletal disorders has been increasing in the last few decades. They currently represent one of the most prevalent causes of absenteeism and disability. According to the WHO, work-related disorders are multifactorial with individual, organizational, psychosocial, and sociocultural factors involved [2,16,20,72]. Thus, numerous risk factors can be associated with the development of musculoskeletal illness, namely long working hours with repetitive tasks, maintained postures, and inadequate working conditions in which to perform the work, including appropriate furniture. Of these factors, a repetition of movements, long hours in the same position, and a lack of pauses are common amongst computer workers [16,72]. Lack of postural variability and the permanence of a sitting position can increase the internal pressure in the nuclei of intervertebral discs, stretch the structures of the spine, and reduce the return circulation of the lower limbs, promoting an inflammatory state in bone and muscle structures with consequent painful symptoms [26,73]. Different studies also reported that long working hours without a pause are risk factors for trigger pain in different segments of the spine such as the cervical or shoulder region [16,74].

Other studies identified the impact of MSS related to work activity in economic terms, namely absenteeism and social spending. For example, it was reported that in Germany MSS and connective tissue disorders accounted for €17.2 billion of lost production in 2016 [75].

In the context of Occupational Health, the reduction of MSS symptoms will be a challenge for the future. In this context, as was previously mentioned, PA-promoting actions within the scope of work activity to reduce MSS will be crucial [2,19,29].

## 4. Discussion 

The WHO’s Model of Healthy Workplaces (Figure 1) advocates that workplaces are privileged places in which to develop health-promotion actions, contributing to a healthy work practice and lifestyle [22,66]. According to this model, a continuous improvement which guarantees that health, safety, and well-being programs are implemented to promote health at work and avoid factors that may create emotional or mental stress and work-related diseases is crucial.

According to the European Agency for Safety and Health at Work (EU-OSHA), the emergence of technologies associated with digital work and the significant increase in the population that works with electronic devices such as computers, either in person or remotely, brings opportunities for workers and employers but also brings new challenges and risks for health and safety at work. Addressing challenges and risks and maximizing opportunities depends on how technologies are applied, managed, and regulated in the contexts of social, political, and economic trends. The main objective is to raise awareness of the impact of new digital technologies at work and in the workplace, of and the challenges and opportunities associated with occupational safety and health. In this context, the European Agency for Safety and Health at Work [76] promotes the Healthy Workplaces 2023–2025 campaign entitled “Safe and healthy work in the digital age” [9].

Health protection and disease prevention initiatives are all vital in the workplace [4]. In that sense, the role played by a physical therapist in the workplace contributes to the physical, functional, and mental health of the worker, which directly influences their quality of life. Therefore, we believe that physical therapists need to know and scientifically understand the impact, benefits, and specific characteristics of an exercise program that intervenes at the physical and mental health level with direct implications for QoL and PA level in CWS.

The adoption of strategies to promote health in the workplace, namely through the implementation of exercise programs, has aroused the interest of professionals from different areas. Multidisciplinary teams including exercise professionals, occupational therapists, and other health professionals such as psychologists and physical therapists play an important part in investigating computer workers’ perception of PA and identifying how to best change attitudes in order to introduce successful physical activity-promotion programs that implement the WHO model of Healthy Workplaces adequately.

Health gains are greater when programs are continuous and oriented towards the specific environment in which participants live and where a large part of personal and/or community problems develop.

## 5. Conclusions

In view of the above, there is evidence that the context of occupational health studies in organizations centered on the education–health–worker relationship should be supported by a paradigm in which workers play an active role in their health plan. To support this, the worker needs to be aware of the existence and scope of the health problems in question. In the workplace, these problems may be the consequences of a given behavior (e.g., occupational stress or overlapping tasks) or the consequences of poor conditions in the workplace (e.g., bad posture or a lack of ergonomic chairs). This knowledge is important because it arouses concern and raises attention, creating an important starting point for change.

In this review, we summarize the evidence indicating that PA is a protective health factor for computer workers and also call attention to the need for committed action on behalf of all intervening in the promotion of the development of an educational program for Occupational Health on different levels including workers’ quality of life, musculoskeletal symptoms, and physical activity. As the WHO healthy work environment model (Figure 1) suggests, there are several factors that can influence the workplace and several steps are necessary to initiate and maintain the health program of each worker, requiring the intervention of health professionals and the active participation of workers in all stages of the process from planning to the final evaluation [66].

## Figures and Tables

**Figure 1 healthcare-10-02457-f001:**
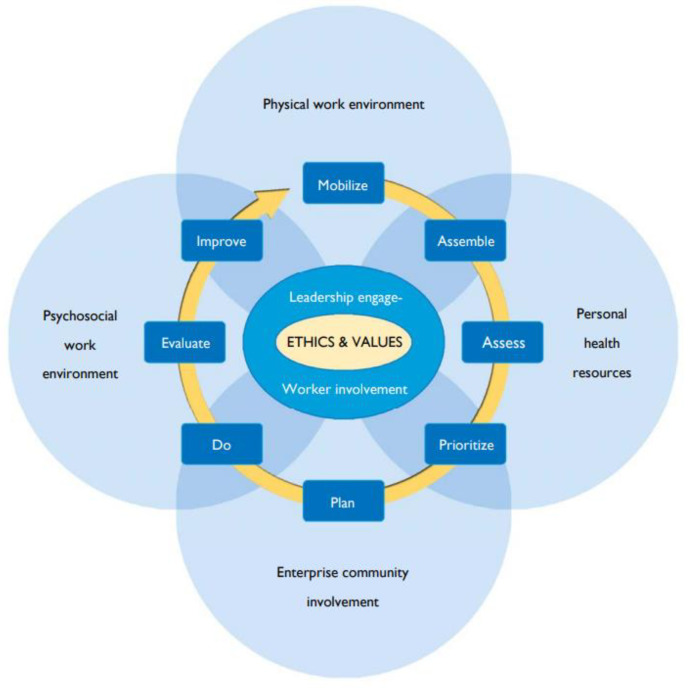
WHO Model of Healthy Workplaces: a model for action [66].

## Data Availability

Not applicable.

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
