# Peer review of "Occupational Health: Physical Activity, Musculoskeletal Symptoms and Quality of Life in Computer Workers: A Narrative Review"

_healthcare, 2022, doi:10.3390/healthcare10122457_

Round 1

Reviewer 1 Report

Dear authors,

Firstly, congratulations on this original and worthy narrative review topic performed. However, based on my point of view, I must regretfully inform you that your manuscript cannot be accepted for publication in the prestigious Healthcare journal.

In this sense, I would like to suggest some aspects to improve the significance of content, and quality of presentation of your results to increase the readers' interest in your study.

First of all, I consider that you have to follow the Healthcare journal guidelines for research manuscripts. I feel a lack of a materials and methods section, explaining the information sources and methods in the process performed in your narrative review such as data searching process, scientific databases names or multi-database searching, search strategies (with inclusion or exclusion criteria), narrative review guidelines used, among others should be explained).

On the other hand, I suggest all the points of your literature review section to show the relationship according to the design of successful health-oriented physical activity promotion actions directed to computer workers. Furthermore, I recommend not duplicating some sentences and paragraphs used in the introduction (e.g.1.sentence from line 254 to 255; e.g.2 sentence from line 260 to 262).

Otherwise, please show the reference cites when you affirm that several authors/works confirm any statement that you show in your manuscript (e.g. line 94, 258, 268) and add references for your affirmations such as for statements of sentences from line 289 to 292; from line 266 to 268. In this way I recommend you review the writing by a native English speaker and modify some words such as “increased immobility” (line 121) for sedentary behavior; “leading to a decrease in” (line 128) for “decreasing; “loss of productivity” (line 303) for “absenteeism” and/or presenteeism, etc….

Furthermore, in the paragraph from lines 260 to 261 you say that it is observed that a large part of the population that has MSS does not perform PA regularly, with the justification of lack of time, work overload and lack of motivation, among others. In this sense, I suggest discussing this statement with this literature review: Calonge-Pascual, S., Casajus-Mallen, J.A., Gonzalez-Gross M. (2020). Adherence Factors Related to Exercise Prescriptions in Health-care Settings: A Review of the Scientific Literature. J Res. Q. Exerc. Sport. doi:https://doi.org/10.1080/02701367.2020.1788699, focused on the analysis of the adherence factors related to physical activity promotion and exercise prescriptions, including patients/workers with musculoskeletal diseases under the WHO five dimensions adherence model.

Finally, on the one hand in your manuscript you show that: “Occupational Health Services is to promote working conditions that guarantee the highest degree of quality of life at work, protecting the health of workers, promoting physical, mental and social well-being and preventing and controlling accidents and diseases by reducing risk factors” and that “…it is extremely important to investigate the PA” “.. and how to change attitudes to introduce successful health-oriented physical activity promotion programmes” and that “the role of the physical therapist becomes essential to improve the quality of life of workers….”. On the other hand, your objective is: “to review the literature to provide a deeper understanding about PA, musculoskeletal symptoms and quality of life in the context of occupational health, contributing to the design of successful health-oriented physical activity promotion actions directed to computer workers.” In this sense, Can you explain Why you suggest constantly only some health professionals such as ergonomists, physiotherapists, or physical therapists (statements from line 212 to 214, or from line 330 to 339) are the best professionals in the implementation of specific physical exercise programmes in an occupational context? I suggest you this reference (McLeroy, K. R., Bibeau, D., Steckler, A., & Glanz, K. (1988). An ecological perspective on health promotion programs J Health Educ Q. 15(4), 351-377. doi:10.1177/109019818801500401) as the possibility to discuss your results by an ecological perspective on health promotion programs and to work them by a multidisciplinary team with other exercise and health personnel such as, exercise professionals, occupational therapists, nurses or sports physicians for instance…. and how important can be the role of psychologists to investigate the perception about PA of computer workers have and how to change attitudes to introduce successful health-oriented physical activity promotion programmes in different social groups and in other to implement the WHO Model of Healthy Workplaces adequately. In this way, you conclude by saying that there are several ways and factors that can influence the workplace and several steps necessary to initiate and maintain the health programme of each worker, requiring the active participation of workers in order to be involved in the stages of the process, from planning to the final evaluation. Please, consider all my reference cites, and suggestions proposed to avoid any kind of bias and improve all your points of view mentioned in your narrative review.

Yours faithfully,

Reviewer 2 Report

This narrative review explains the relationship between physical activity, musculoskeletal symptoms and quality of life in computer workers. The topic is relevant and the context of teleworking was well addressed. I have some suggestions:

1. Reviewing the title, it is not clear why the authors used the term "health education";

2. Objective: it is not clear how the results will contribute to the design of successful health promotion actions;

3. Results: the inclusion of more intervention studies can bring more insights to the practitioners;

4. Conclusions: Specify/interpret the WHO Healthy Workplaces Model.

Round 2

Reviewer 1 Report

Dear authors,

Congratulations, I consider that all my comments and suggestions had been answered point-by-point. 

I accept the manuscript in this present form to be published.

Yours faithfully.